# Comparison of UESCOPE VL 400, I-View, Non-Channeled Airtraq Videolaryngoscopes and Macintosh Laryngoscope for Tracheal Intubation in Simulated Out-of-Hospital Conditions: A Randomized Crossover Manikin Study

**DOI:** 10.3390/healthcare12040452

**Published:** 2024-02-10

**Authors:** Paweł Ratajczyk, Przemyslaw Dolder, Bartosz Szmyd, Manuel A. Gomez-Rios, Piotr Hogendorf, Adam Durczyński, Tomasz Gaszyński

**Affiliations:** 1Department of Anesthesiology and Intensive Therapy, Medical University of Lodz, 90-419 Lodz, Poland; przemyslaw.dolder@umed.lodz.pl; 2Department of Neurosurgery and Neuro-Oncology, Medical University of Lodz, 90-419 Lodz, Poland; bartoszmyd@gmail.com; 3Department of Anesthesiology and Perioperative Medicine, Complejo Hospitalario Universitario de A Coruña, 15006 A Coruña, Spain; magoris@hotmail.com; 4Anesthesiology, Perioperative Medicine and Pain Management Research Group, 15006 A Coruña, Spain; 5Spanish Difficult Airway Group (GEVAD), 15006 A Coruña, Spain; 6Department of General and Transplant Surgery, Medical University of Lodz, 90-419 Lodz, Poland; piotr.hogendorf@umed.lodz.pl (P.H.); adam.durczynski@umed.lodz.pl (A.D.)

**Keywords:** videolaryngoscopes, intubation success, simulation, novices, pre-hospital conditions

## Abstract

The aim of the study was to test the hypothesis that the results obtained with three different types of video laryngoscopes (UESCOPE VL-400, I-View, Non-Channeled Aitraq) with and without an endotracheal stylet should be better than the results obtained with a Macintosh laryngoscope in a simulated out-of-hospital scenario by a person without clinical experience. Primary outcome measures were the time taken to successfully achieve tracheal intubation (TI). Secondary outcomes included the grade of glottic view (Cormack and Lehane grades 1–4), the incidence of successful TI, the number of audible dental clicks indicating potential dental damage, the level of effort required to perform TI, and the operator’s comfort during the procedure. The time required to achieve tracheal intubation successfully was significantly longer with the Macintosh laryngoscope and Airtraq than with the other video laryngoscopes. The use of the stylet significantly reduced the time required for tracheal intubation with the Macintosh laryngoscope (21.8 sec. vs. 24.0 sec., *p* = 0.026), UESCOPE VL 400 (18.1 sec. vs. 23.4 sec., *p* = 0.013), and Airtraq (22.7 sec. vs. 34.5 sec., *p* < 0.001). There were no significant differences in intubation time when using the I-View with or without stylets. No differences were observed in the Cormack–Lehane grading. The success rate of intubation was 100% for the Macintosh and I-View laryngoscopes used with or without stylets and for the UESCOPE VL 400 and Airtraq laryngoscopes used with stylets. Without stylets, the success rate of intubation was 96.6% for the UESCOPE VL 400 and 86.6% for the Airtraq. There were no significant differences in the risk of dental damage between the Macintosh, UESCOPE VL 400, I-View, and Airtraq laryngoscopes, regardless of the use of stylets (without and with stylets). The use of stylets significantly reduced dental damage only for the Airtraq laryngoscope: 8 (26.6%) vs. 2 (6.6%). Statistically significant differences in perceived exertion were observed between the mentioned laryngoscopes, both with and without stylets. However, there were no differences in the comfort of use between the laryngoscopes, regardless of the use of stylets (without and with stylets. The use of stylets led to better comfort in the case of the Macintosh (2.5 vs. 3, *p* = 0.043) and UESCOPE VL 400 (2 vs. 3, *p* = 0.008) laryngoscopes. In our study, the I-View and UESCOPE VL-400 video laryngoscopes provided better intubation results than the Macintosh laryngoscope in terms of time needed to intubate, glottis visibility, and reduction in dental damage. The use of the stylet did not significantly improve the intubation results compared to the results obtained in direct laryngoscopy. Due to the small study group and the manikin model, additional studies should be performed on a larger study group.

## 1. Introduction

Tracheal intubation (TI) is a vital and potentially life-saving procedure that is commonly performed in emergency medicine [1]. It is often required for patients who are unable to maintain adequate oxygenation or ventilation, such as those with trauma, severe respiratory distress, or cardiac arrest. Difficult or unsuccessful intubation can lead to severe consequences [2], particularly for inexperienced rescuers [3]. Multiple attempts at intubation have been proven to be associated with an increased incidence of adverse events, delayed transport, prolonged hospitalization, worse neurological outcomes, and increased mortality [4,5]. In the prehospital setting, intubation can be particularly challenging, especially when the patient is lying on the ground, as it can restrict the ability of the healthcare provider to access the airway [6].

In recent years, video laryngoscopy has emerged as a promising tool for improving intubation success rates, especially in difficult airway scenarios. It offers superior visualization of the larynx, improved success rates in patients with difficult airways, and lower incidence of complications compared to the Macintosh laryngoscope [7]. They also make teaching and learning intubation easier, with shared views between teachers and students, and the ability to provide feedback [8]. However, the large number of video laryngoscopes available on the market can make it difficult for those with little or no clinical experience to use them effectively. However, the use of video laryngoscopy in prehospital tracheal intubation with patients on the ground has not been extensively studied [9].

Airway management in emergencies requires both theoretical knowledge and manual skills [3]. In prehospital care, the success of intubation depends not only on the type of device used, but also on the operator’s training and experience. Therefore, it is crucial to look for user-friendly devices that result in more-effective intubation while introducing additional techniques to reduce the risk of complications. Simulation training is a valuable tool for healthcare providers to practice and improve their skills in a safe and controlled environment. It has been shown to enhance clinical performance, reduce errors, and increase patient safety. The use of simulation training for out-of-hospital tracheal intubation is particularly useful, as it allows healthcare providers to practice in a realistic setting without risking harm to actual patients.

We hypothesize that the outcomes obtained with three different types of video laryngoscopes with and without the combined use of an intubation stylet should be superior to those obtained with direct laryngoscopy in a simulated out-of-hospital scenario with a patient on the ground and the operator without clinical experience. The primary is the time taken to successfully achieve TI with each device. Secondary endpoints included the grade of glottic view, the incidence of successful TI, the number of audible dental clicks, the level of effort required to perform TI, and the operator’s comfort during the procedure.

## 2. Methods

This study was conducted at the Norbert Barlicki University Clinical Hospital No. 1 in Lodz, Poland, from 28 February 2021 to 30 June 2021, following approval from the Medical University of Lodz Bioethics Committee (ref: RNN/363/13/KB, date: 20 February 2021). Prior to the start of recruitment, the study was registered on ClinicalTrials.gov (NCT number: NCT05946694). All participants provided informed written consent for voluntary participation. Students who met the inclusion criteria were invited to participate in the study, which included being a third-year full-time first-degree student in Emergency Medicine at the Medical University of Lodz and willing to participate after being randomly selected according to the template number. The exclusion criterion was previous clinical experience with the laryngoscopes used in the study.

This trial was a simulation-based, single-center, randomized, crossover study.

### 2.1. Pre-Study Preparation

All participants attended a 30 min lecture on laryngoscope construction, the principles of usage, the anatomical structure of the airways, and intubation techniques based on manufacturers’ recommendations. Following the lecture, the instructor demonstrated proper intubation using each of the four laryngoscopes that were being tested. Next, the students participated in a supervised workshop where they had the opportunity to intubate a manikin of head, neck, and lower airways placed on an operating table at an optimal height, with the manikin’s head in a neutral position, using each of the tested laryngoscopes with and without a stylet. One month later, 30 students participated in the actual study.

### 2.2. Simulation Model

To create simulated out-of-hospital conditions, a certified airway training manikin (Laerdal Airway Management Trainer Stavanger Norway of universal difficulty) was placed in a neutral position at the floor level.

The trial tested three non-channeled video laryngoscopes: UESCOPE VL 400 (Zhejiang UE Medical Corp. Xianju China), I-View™ VL (Intersurgical Ltd., Wokingham, Berkshire, UK), Airtraq sp ^®^ (Prodol Meditec S.A, Vizcaya, Spain), and a Macintosh laryngoscope (HEINE Optotechnik GmbH & Co. KG, Gilching, Germany) (see Figure 1).

Each participant performed eight intubations on the manikin using all four devices. The intubation order was randomized using a blocked randomization strategy generated by the Randomizer Program (randomizer.org, accessed on 14 January 2024). A sealed opaque envelope was used to determine the order in which the laryngoscopes were used and whether the tube was with or without a stylet.

After each intubation, there was a minimum 10 min pause before the next one. The subject then proceeded to the next intubation with the randomly selected laryngoscope, with the tube and stylet as determined by the sealed envelope.

All intubations were performed using a polyvinyl chloride cuffed endotracheal tube (Covidien LLC, Hampshire Street, Mansfield, MA, USA; internal diameter 7.0 mm) and an endotracheal stylet (Sumi^®^ from sp. z o.o., 35 Drobiarska Street, 05-070 Sulejówek, Poland), when used. The endotracheal tubes and stylets were coated with a standard lubricant for simulators. All data were pseudonymized for each simulation.

### 2.3. Outcomes

Primary outcome measures were the time taken to successfully achieve tracheal intubation (TI). Secondary outcomes included the grade of glottic view (Cormack and Lehane grades 1–4), the incidence of successful TI, the number of audible dental clicks indicating potential dental damage, the level of effort required to perform TI, and the operator’s comfort during the procedure.

The time was measured by the same investigator (P.R.) using a stopwatch for all TI attempts. The timing measurement started when the participant picked up the laryngoscope and ended when initial ventilation was achieved using a resuscitation bag after the placement and sealing of the endotracheal tube. Successful TI was confirmed by observing the breathing movements of the manikin lungs. An unsuccessful attempt was defined by the absence of respiratory movements of the dummy or an intubation time of more than 60 s. This criterion was adopted because the study aimed to assess the suitability of the devices for individuals without clinical experience in intubation. Participants without sufficient skills or a good view of the glottis could prolong intubations without knowing how to solve the problem. In a real clinical scenario, such prolongation of intubation attempts beyond 60 s could result in irreversible brain damage in the victim, especially considering the necessary time for the emergency medical team to arrive.

Participants rated the level of effort required to perform TI using the modified Borg scale (0—no effort, 10—maximum effort) and graded their comfort during the procedure as “full comfort, satisfactory comfort, moderate comfort, unsatisfactory comfort, and no comfort”.

### 2.4. Statistical Analysis

We used Statistica 13.1PL (StatSoft, Poland, Krakow) to perform the statistical analysis.

The distribution of the continuous data was checked with the Shapiro–Wilk test. The majority of the data had a distribution other than normal (*p* < 0.05) or were measured on the ordinal scale. Therefore, they are further presented as the median with the interquartile range (IQR). The dependencies between them were assessed with Kruskal–Wallis’s test, Dunn’s post hoc tests for independent data, and with Wilcoxon’s test for dependent data. Nominal data are presented as n (%) and were assessed with a test chosen based on the size of the smallest subgroup: n < 5—Fisher’s exact test, 5 ≤ n < 15—Yates’s chi-squared test, and 15 < n—chi-squared test.

The sample size was calculated using the R Stats Package v4.2.2. Based on previous student training registers, we would consider 30 s to be a significant difference between means, so 30 participants would yield a type-1 error of 5% and a power of 90%.

## 3. Results

### 3.1. Participants’ Characteristics

Thirty participants were enrolled in the study and were randomized. All of them completed the study as indicated in the flow chart (Figure 2). Of the participants, 18 (60%) were female and 12 (40%) were male, with a mean age of 22 (SD) years. Prior to the study, all participants had only used the Macintosh laryngoscope for intubation. Among the participants, 14 students had experience with less than 10 prior intubations, 10 had performed between 10 and 20 intubations, and 6 had performed more than 20 intubations on a manikin. No participant had an experience of more than 50 intubations on a manikin model nor intubated a real patient.

### 3.2. Primary Endpoints

The time required to achieve tracheal intubation successfully was significantly longer with the Macintosh laryngoscope and Airtraq than with the other video laryngoscopes.

The use of a stylet significantly reduced the time required for tracheal intubation with the Macintosh laryngoscope (21.8 sec. (IQR: 17.2–26.7) vs. 24.0 sec. (IQR: 19.4–36.0), *p* = 0.026), UESCOPE VL 400 (18.1 sec. (IQR: 14.3–25.1) vs. 23.4 sec. (IQR: 16.3–37.6), *p* = 0.013), and Airtraq (22.7 sec. (IQR: 16.4–26.1) vs. 34.5 sec. (IQR: 27.6–39.1), *p* < 0.001). There were no significant differences in intubation time when using the I-View without or with stylets (16.6 sec. (IQR: 13.9–21.2) vs. 20.3 sec (IQR: 14.9–24.6), *p* = 0.213) (Figure 3).

As the *p*-value was less than 0.001 in the Kruskal–Wallis test, we conducted a post hoc Dunn test, and the results are presented in Figure 3.

### 3.3. Secondary Endpoints

No differences were observed in the Cormack–Lehane grading (see Table 1).

The success rate of intubation was 100% for the Macintosh and I-View laryngoscopes used with or without stylets and for the UESCOPE VL 400 and Airtraq laryngoscopes used with stylets. Without stylets, the success rate of intubation was 96.67% (*p* = 1.000) for the UESCOPE VL 400 and 86.67% (*p* = 0.056) for the Airtraq (Table 1).

There were no significant differences in the risk of dental damage between the Macintosh, UESCOPE VL 400, I-View, and Airtraq laryngoscopes, regardless of the use of stylets (*p* = 0.241 and *p* = 0.165, without and with stylets, respectively; see Table 1). The use of stylets significantly reduced dental damage only for the Airtraq laryngoscope: 8 (26.67%) vs. 2 (6.67%).

Statistically significant differences in perceived exertion were observed between the mentioned laryngoscopes, both with (*p* < 0.001) and without stylets (*p* < 0.001). The exact *p*-values of Dunn’s post hoc test are shown in Figure 4A,B. However, there were no differences in the comfort of use between the laryngoscopes, regardless of the use of stylets (*p* = 0.240 and *p* = 0.132, without and with stylets, respectively; see Table 1). The use of stylets led to better comfort in the case of the Macintosh (2.5 (IQR: 2–3) vs. 3 (IQR: 2–3), *p* = 0.043) and UESCOPE VL 400 (2 (IQR: 2–3) vs. 3 (IQR: 2–3), *p* = 0.008) laryngoscopes (Table 1).

## 4. Discussion

This simulation-based, randomized, crossover study trial showed that the I-View laryngoscope proved to be the best-rated laryngoscope in our study, with respect to the primary and secondary endpoints of the study. It showed a 100% success rate for intubation without a stylet and with a stylet, the fastest intubation times in both cases being, respectively: 16.6 s vs. 20.3 s. It is the only laryngoscope in our study that showed shorter intubation times without a stylet than with one. For secondary endpoints, it also proved to be similar to the UESCOPE in terms of comfort and intubation effort. The percentage of tooth damage in our study was 13.3% regardless of whether we used a stylet or not. In the absence of a stylet, this percentage was the same as for the least-traumatic laryngoscope in our study, which was the UESCOPE VL 400. In the situation of using a stylet, the UESCOPE VL 400 and Airtraq laryngoscopes, where the percentage of tooth damage was 3.3% and 6.6%, respectively, proved to be the safer devices. In the case of this laryngoscope, the use of a stylet statistically significantly increased the comfort of intubation compared to the variant without a stylet, resulting in a lack of negative evaluations associated with its use. This laryngoscope had the best view of the glottis according to the Cormack–Lehane scale.

In simulated out-of-hospital conditions, where intubation of the manikin took place at the floor level, the absence of the need to maintain the eye–glottis line is important, as it does not require the intubator to adopt a more-forced, bent body position, which is uncomfortable and non-ergonomic [3]. In the case of a video laryngoscope such as the I-View laryngoscope, the ability to assess the view of the glottis through the device’s monitor makes the adopted body position less crouched and more user friendly to the intubator [3]. This is important in the situation of intubating a patient by those without experience in airway management, where, in the case of the possibility of choosing between the Macintosh laryngoscope and video laryngoscopes including the I-View, some authors suggest choosing the latter [10]. In the case of intubation by anesthesiologists, Wakabayashi believes that, despite the easier handling and better glottal visibility provided by video laryngoscopes, the efficiency and intubation times with the classic Macintosh laryngoscope are at an acceptable level, comparable to the use of the I-View laryngoscope. This is crucial given the widespread availability of Macintosh laryngoscopes and the still limited video laryngoscopes [11]. Of the video laryngoscopes, some authors suggest that the I-View laryngoscope is the appropriate device for use in difficult prehospital care settings, due to its ease and disposability [12]. In their study, Maritz et al. showed that the use of video laryngoscopes provided better intubation conditions, allowed better visualization of the glottis, and thus, facilitated intubation when used by anesthesiologists with extensive experience in conventional and video laryngoscopy and paramedics with little prior experience in conventional and no experience in video laryngoscopy [13]. Although the use of video laryngoscopes did not affect intubation success among anesthesiologists, in the hands of paramedics with little intubation experience, it reduced the failure rate from 14.8% with a conventional Macintosh laryngoscope to 3.7% with video laryngoscopes [14]. Similar conclusions were reached in the study by Toshiyuki, who found that, for anesthesiologists, it did not matter whether one used an I-View video laryngoscope or a Macintosh laryngoscope; the intubation times were similar, but the former proved better at visualizing the glottis based on the Cormack–Lehane scale; it was easier and more user-friendly than a Macintosh laryngoscope [10].

According to some authors, the I-View Laryngoscope improves glottic visualization in patients with morbid obesity, which is one of the risk factors for difficult intubation. In his study, Gaszynski showed that the POGO laryngoscopy score (scale of assessment of visibility of the glottis based on the percentage of glottis opening) was better for the McGrath Mac video laryngoscope than for the I-View laryngoscope, but both devices allowed safe and effective intubation in overweight patients. The hemodynamic response to video laryngoscopy in this study group was similar between devices [14].

In our study, for the UESCOPE VL 400 laryngoscope, the use of a stylet resulted in an increase in the percentage of successful intubations from 96.6% in the situation without a stylet to 100% when one was used. This laryngoscope, like the I-View laryngoscope, had an intubation success rate of 100% when a stylet was used. The average intubation times were slightly longer than those of the I-View laryngoscope i.e., 23.4 s and 18.1 s for the no-stylet and stylet scenarios, respectively, statistically significantly reducing this time. The use of this laryngoscope was associated with the same intubation effort as the I-View laryngoscope in the absence of a stylet and slightly less when a stylet was used. The use of a stylet with this laryngoscope statistically significantly increased the intubation comfort compared to the variant without a stylet. In the case of the UESCOPE VL 400 laryngoscope, the use of the stylet resulted in a reduction in the percentage of tooth damage from 13.3% to 3.3%. The glottis visibility obtained with this laryngoscope was only slightly worse than that obtained when using the I-View laryngoscope.

Knowing the distinctiveness of using the UESCOPE VL 400 laryngoscope is crucial before using it, hence the probably slightly worse performance of this device, in our study, compared to the I-View laryngoscope. A common mistake is to place the blade tip of this laryngoscope too close to the glottis, especially for novice intubators. Both withdrawing it slightly and providing a “bird’s-eye” view provides more space for the endotracheal tube to pass and reduce the angle at which the endotracheal tube must pass [8].

Because the endotracheal tube must be inserted “around the corner”, some authors recommend using a stylet to match the shape of the endotracheal tube to the curvature of the blade. In adult patients with normal airways, Gao et al. showed that both 40° and 60° angles of the endotracheal tube with a stylet can ensure successful intubation with the UESCOPE VL 400 laryngoscope, but a stylet bent at 60° results in a higher percentage of successful first intubation and a shorter intubation time [8]. Similar data were obtained in our study where the use of a stylet increased the percentage of successful intubations and had a positive effect on intubation comfort and reduced the percentage of dental injuries.

In a large randomized controlled trial involving 600 adult patients, Wang et al., comparing the use of the UESCOPE VL 400 and Macintosh laryngoscope, showed that the UESCOPE provided a better view of the larynx, a higher rate of successful intubation, and a shorter time required for laryngeal visualization and intubation [15]. Both laryngoscopes were comparable in terms of intubation complications. Other studies in patients with normal airways have shown that glottis visualization was improved, and the need to compress the thyroid cartilage during laryngoscopy, as well as the time required to visualize and intubate the larynx, was reduced with the UESCOPE VL 400 laryngoscope compared to the Macintosh laryngoscope [16,17]. Similar conclusions were reached by Yang and Sun, who found that, although the laryngeal view and intubation success rate were not significantly different between the UESCOPE VL 400 and Macintosh laryngoscopes, the intubation time was significantly shorter and the sore throat significantly lower when using the UESCOPE VL 400 [18,19]. Management of a difficult airway is always a challenge for the intubator. An et al. in this group of patients showed that the UESCOPE VL 400 laryngoscope provided a better view of the larynx, a higher intubation success rate, an easier intubation technique, and a weaker hemodynamic response than the Macintosh and Glidescope laryngoscopes, but resulted in a longer time required to visualize the larynx [20]. However, another clinical study found that intubation time was reduced with the UESCOPE VL 400 laryngoscope in this group of subjects [8]. In their study of obese patients, Wang and Zhang showed that intubation efficiency increased and the rate of airway complications decreased with the UESCOPE VL 400 laryngoscope compared to the Macintosh laryngoscope, but the hemodynamic response was not significantly different between the two devices [21]. Similar results were obtained by Li et al. in their study, but in this case, the study group was patients with post-burn neck and facial contractures [22].

In a study involving 265 patients requiring intubation in the emergency department, Wang et al. compared the efficacy and safety of the UESCOPE and Macintosh laryngoscope when intubation was performed by emergency physicians. For patients with a grade 3 or 4 glottis view according to the Cormack–Lehane scale, the use of the UESCOPE VL 400 laryngoscope was associated with a significant reduction in the time to visualization and intubation of the larynx, an increase in the percentage of correct intubation, and a decrease in the incidence of airway trauma complications and unintentional esophageal intubation [20].

Pan et al. concluded that the UESCOPE VL 400 laryngoscope can improve the intubation skills of untrained emergency physicians. They showed that, for emergency intubation with the UESCOPE laryngoscope, differences in the percentage of successful intubation, an increased rate of esophageal intubation, an increased intubation time, and an increased rate of intubation complications were not statistically significant between untrained and experienced clinicians [23].

In our study, the data obtained put the UESCOPE VL-400 laryngoscope on a par with the I-View laryngoscope and were better than the results of the Airtraq video laryngoscope and the reference Macintosh laryngoscope.

The UESCOPE VL 400 laryngoscope was found to be a more-useful device among those without clinical experience than the Airtraq laryngoscope. In the available literature comparing the UESCOPE VL 400 laryngoscope and the Airtraq laryngoscope, studies conducted in patients with normal airways showed that the glottal image, intubation time, and intubation efficiency did not differ significantly between these laryngoscopes, but these studies evaluated the usefulness of these laryngoscopes on a different study group than our study [8].

For the Airtraq laryngoscope, the use of a stylet led to an increase in the percentage of successful intubation from 86.67% to 100% and a statistically significant decrease in intubation time from 34.52 s to 22.75 s. Compared to the other laryngoscopes used in the study, the average intubation times were the longest for the Airtraq laryngoscope for both the scenarios without and with the stylet. However, they were within acceptable values for laryngoscopes. For secondary endpoints, the use of a stylet resulted in a statistically significant reduction in the percentage of tooth damage from 26.67% to 6.67% and did not affect intubation comfort. A big difference in these statistics in comparison to other laryngoscopes may stem from the different curvature of a blade, demanding an altered intubation technique, making it much harder for inexperienced intubators to use it without a stylet. Inexperienced intubators tend to prefer learning the most-wildly used Macintosh-type laryngoscopes and videolaryngoscopes, and any other laryngoscope requiring another technique is a difficulty for them. This laryngoscope had the highest effort required to intubate the manikin of all laryngoscopes regardless of the variant with and without the stylet. The use of the Airtraq laryngoscope did not significantly improve the visibility of the glottis compared to the other laryngoscopes included in the study.

Failed tracheal intubation is one of the important causes of morbidity and mortality in susceptible patients [16,17]. Nearly 30% of anesthesia-related deaths are due to the complications of difficult intubation, and more than 85% of all patient-oxygenation complications result in brain damage or death [24]. The antidote to difficult airway problems was supposed to be the Airtraq. In our study, it proved to be the least useful for those without clinical experience. According to Saracoglu, the undeniable advantage of the Airtraq laryngoscope is that it can be used in patients with limited cervical spine mobility and a small degree of oral dilation (provided it is greater than 3 cm). The advantage of the use of this device decreases when we have to deal with secretions and soft tissue swelling. This is not a laryngoscope whose use is intuitive. Proficient use of it requires time and experience [25]. Giquello also came to similar conclusions. In his study, the average intubation time using the Airtraq laryngoscope was about 47 s, and the intubation success rate oscillated at 80% [25]. A slightly higher intubation success rate of 85% was reported by Kleine-Brueggeney [26]. Similar results were obtained in our study, where, as in Giquello’s study, the study group consisted of subjects with no experience in using the Airtraq laryngoscope. A separate conclusion was reached by Lu, who, based on a review of the available literature, concluded that, for those with little or no experience, the use of the Airtraq laryngoscope facilitates faster and more-accurate intubation, reducing the risk of the placement of the endotracheal tube in the esophagus [27]. According to Nowicki, the Airtraq laryngoscope provides as fast or even a faster intubation as the Macintosh laryngoscope, reduces the number of necessary maneuvers to optimize the view of the airway entrance, and lowers the risk of tooth damage. According to him, the learning curve is short, and this is true under normal, as well as difficult airway conditions [28]. Timmel, on the other hand, believes that the Airtraq laryngoscope cannot be recommended as a primary airway device in the prehospital setting without the extensive prior clinical experience gained in the operating room. This is because he believes that the process of learning the clinical use of the Airtraq laryngoscope is much longer than described in the anesthesiology literature [29]. According to Wetsch, video laryngoscopes used by experienced anesthesiologists did not facilitate endotracheal intubation in a model with an immobilized cervical spine any faster or safer than conventional laryngoscopy. It was the Macintosh laryngoscope that provided faster intubation than the Airtraq, although the intubation was faster with the latter than with other video laryngoscopes. In our study, where the study group was people with no clinical experience, the intubation time for the Airtraq was the longest [30].

In our study, the Macintosh laryngoscope showed the same intubation efficiency of 100% for variants with and without a stylet as the I-View video laryngoscope. However, it was inferior to it in terms of the intubation time of 24.0 s in this case vs. 16.6 s in the situation without a stylet and 21.8 s vs. 20.3 s when a stylet was used. In the case of this laryngoscope, the use of a stylet resulted in a statistically significant reduction in intubation time, better intubation comfort, and a reduction in the percentage of tooth damage from 30% to 20%. The Macintosh laryngoscope had a similar intubation effort as the use of the Airtraq laryngoscope, and it was significantly higher than when using the UESCOPE VL 400 and I-View video laryngoscopes. The Cormack–Lehane scale view of airway entry that could be obtained using the Macintosh laryngoscope was the worst of all the devices involved in our study, but there was no statistical difference.

Different opinions are found in the literature regarding clinical situations in which a Macintosh laryngoscope is superior or inferior to video laryngoscopes or other types of laryngoscopes based on direct laryngoscopy. An important piece of information that affects the evaluation of the use of a particular type of laryngoscope is who is ultimately expected to use the type of device. According to many authors, in the situation of a lack of clinical experience and anticipated occasional use of a given laryngoscope, it is definitely important to use such a device, which is characterized by relatively high simplicity and commonness of use. In this situation, the classic Macintosh laryngoscope is of primary importance, regardless of whether we are dealing with normal or difficult airways [31,32,33,34]. For investigators with extensive clinical experience in intubation, who perform hundreds or even thousands of intubations per year, the role of video laryngoscopes, especially in patients with difficult airways, definitely increases. According to Wetsch, who studied the use of Macintosh laryngoscopes and video laryngoscopes in trapped car accident victims, the latter provide a better view of the glottis, but there is some significant delay of intubation. As in our study, he concluded that the use of a stylet allows for faster intubation and makes the method have less failure. He also showed that, under these conditions, no video laryngoscope is superior to direct laryngoscopy with a Macintosh laryngoscope [28].

According to Hansel, who compared video laryngoscopes with the classic Macintosh laryngoscope based on a review of the literature, video laryngoscopes likely reduce the rates of failed intubations and hypoxemia. The devices may also increase success rates on the first intubation attempt and likely improve the visualization of the glottis when assessed as Cormack–Lehane grade 3 and 4. He also found a difference in esophageal intubation rates, but this finding was supported by low-confidence evidence [35]. It is also impossible to show which type of laryngoscope causes less dental trauma. The author concluded that video laryngoscopy probably provides a safer risk profile compared with classic laryngoscopy using a Macintosh laryngoscope. According to Higazawa, the Macintosh laryngoscope should be the one on which inexperienced medical students should learn intubation, as all other laryngoscopes are more difficult to use and often require a long learning period [16]. When using videolaryngoscopes, several studies have confirmed that Macintosh-like blades are easier to use [36,37,38] and may reduce the necessity to use a stylet during intubation efforts [39,40]. When comparing the Macintosh-like blades with hyperangulated videolaryngoscopes, it seems that the Macintosh-like blades VL are better to learn in both easy and difficult airway scenarios for unexperienced providers.

A study performed showed that it is necessary to constantly practice airway management methods including performing endotracheal intubation [41]. It is particularly important to learn how to use multiple laryngoscopes, as this can be useful in unconventional situations that require the modification of the technique, equipment, or body position [33]. This is because any exercise in this area reduces the risk of error, reduces the stress of those performing the procedure, and, most importantly, increases the chances of the victim’s survival and recovery [34].

## 5. Conclusions

In our study, the I-View and UESCOPE VL-400 video laryngoscopes provided better intubation results than the Macintosh laryngoscope. The use of the stylet did not significantly improve the intubation results compared to the results obtained in direct laryngoscopy. Due to the small study group and the manikin model, additional studies should be performed on a larger study group.

### Limitations

The study has several limitations. It was conducted on a manikin model, where simulated out-of-hospital conditions were created by placing the manikin at the floor level, without the influence of other external factors affecting the effectiveness of intubation. Difficult airway scenarios were also not tested. The study group was relatively small and consisted of Emergency Medicine students who had little prior experience intubating a manikin with a Macintosh laryngoscope.

## Figures and Tables

**Figure 1 healthcare-12-00452-f001:**
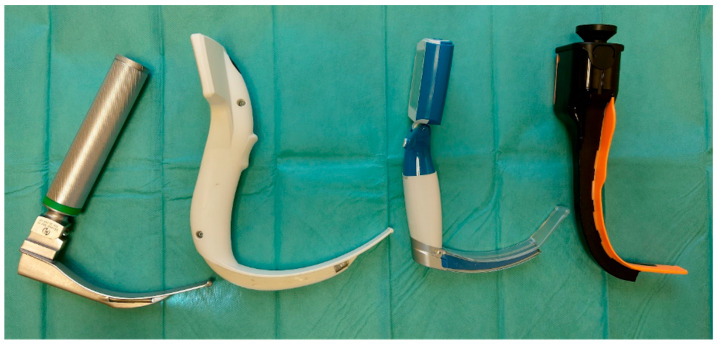
From the left: Macintosh laryngoscope, I-View laryngoscope, UESCOPE VL 400 laryngoscope, Non-Channeled Airtraq laryngoscope.

**Figure 2 healthcare-12-00452-f002:**
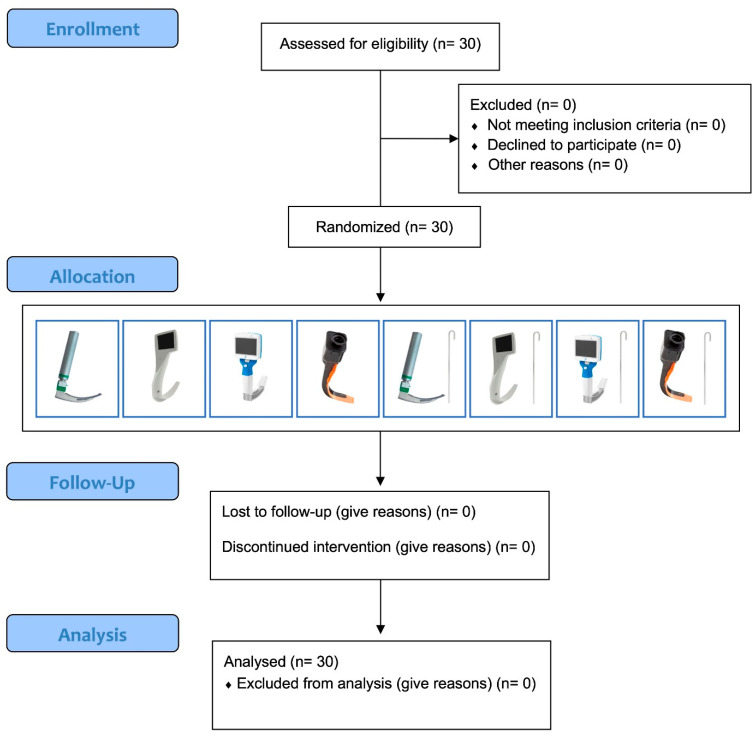
Flow chart. Each participant performed intubation in all settings in a randomized controlled order. There were no drop-outs.

**Figure 3 healthcare-12-00452-f003:**
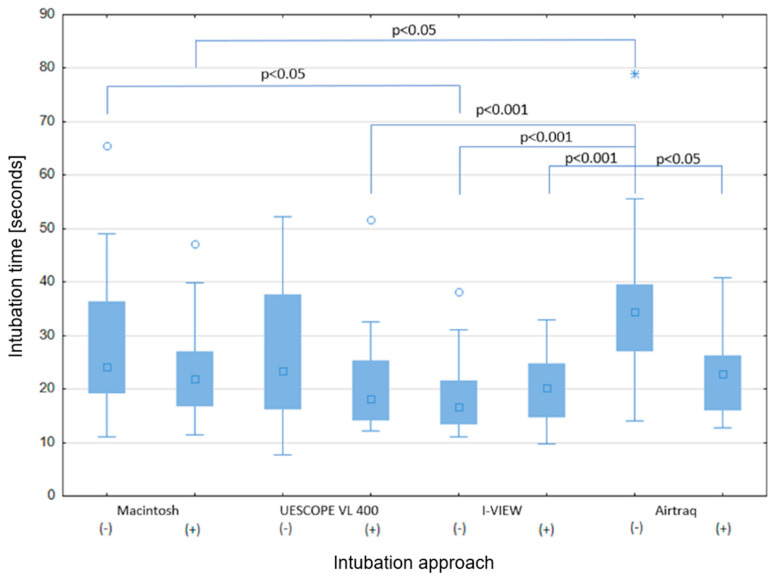
The median time of intubation for different intubation approaches (Kruskal–Wallis test: *p* < 0.001, the presented *p* was taken from the Dunn test.

**Figure 4 healthcare-12-00452-f004:**
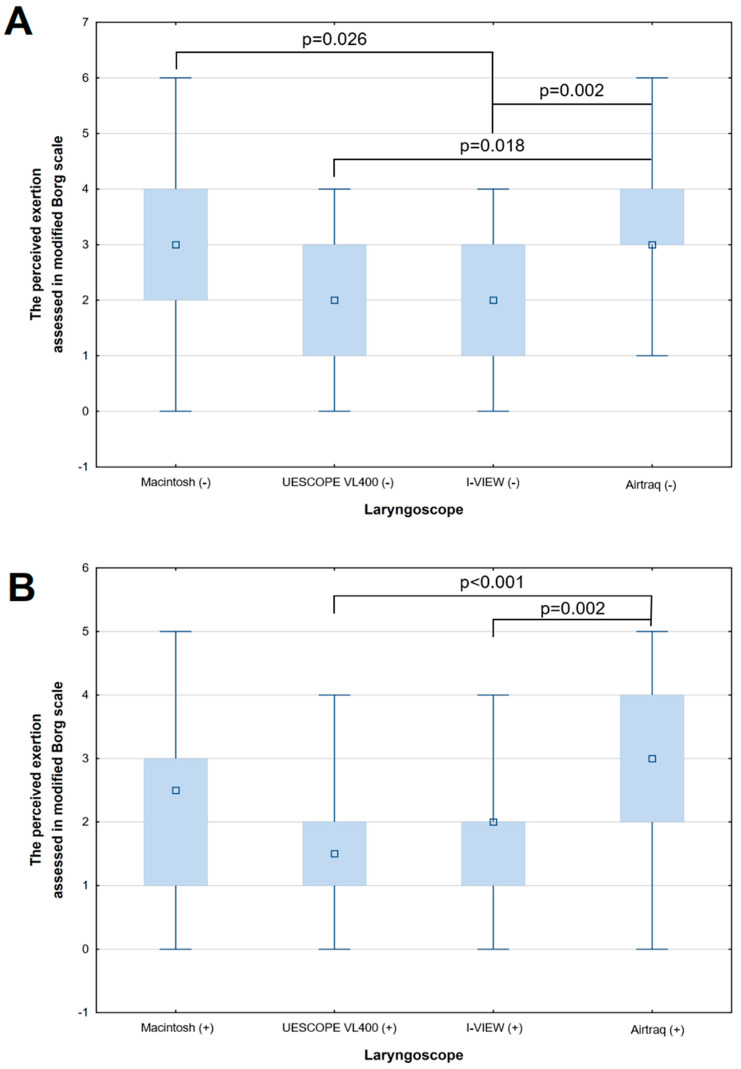
The comparison between the perceived exertion between the four laryngoscopes (Macintosh, UESCOPE, I-View, and Airtraq) used in both manners: (**A**) without (*p* < 0.001 in Kruskal–Wallis Test) and (**B**) with stylets (*p* < 0.001 in Kruskal–Wallis Test).

**Table 1 healthcare-12-00452-t001:** Primary and secondary study outcomes n (%, CI 95%) or median (IQR (range)).

	UESCOPE VL 400 (n = 30)	I-View (n = 30)	Airtraq (n = 30)	Macintosh (n = 30)	*p*-Value
**Primary Outcome**					
Time to TI; s					
*without stylet*	23.31 (IQR: 17–31)	16.6 (IQR:13–22)	34.5 (IQR: 28–39)	24.1 (IQR:19–37)	<0.05
*with stylet*	18.18 (IQR: 14–25)	20.3 (IQR:15–24)	22.7 (IQR: 17–27)	18.5 (IQR:17–28)	>0.05
**Secondary Outcomes**					
Cormack-Lehane grading					
*without and with stylet*					
1	31 (51.7%)	31 (51.7%)	27 (45%)	26 (43%)	>0.05
2	26 (43.7%	27 (45%)	28 (46.7%)	24 (40%)	>0.05
3	3 (5%)	2 (3.3%)	5 (8.3)	9 (15%)	<0.05
4	0 (0%)	0 (0%)	0 (0%)	1 (1.7%)	>0.05
Successful TI rate					
*without stylet*	29 (96.7%)	30 (100%)	26 (86.7%)	30 (100%)	>0.05
*with stylet*	30 (100%)	30 (100%)	30 (100%)	30 (100%)	>0.05
Dental clicks					
*without stylet*	4 (13.3%)	4 (13.3%)	8 (26.7%)	9 (30%)	<0.05
*with stylet*	1 (3.3%)	4 (13.3%)	2 (6.7%)	6 (20%)	<0.05
Effort					
*without stylet*	2 (IQR: 1–3)	2 (IQR: 1–3)	3 (IQR: 3–4)	3 (IQR: 2–4)	>0.05
*with stylet*	2.5 (IQR: 1–3)	2 (IQR: 1–2)	3 (IQR: 2–4)	2.5(IQR:1–3)	>0.05
Comfort					
*without stylet*	2 (IQR: 2–3)	3 (IQR: 2–3)	2 (IQR:1.25–3)	2.5(IQR:2–3)	>0.05
*with stylet*	3 (IQR: 2–3)	3 (IQR: 2–3)	2 (IQR:2–3)	3 (IQR: 2–3)	>0.05

TI—tracheal intubation.

## Data Availability

Data are contained within the article.

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
