# Peer review of "Comparison of UESCOPE VL 400, I-View, Non-Channeled Airtraq Videolaryngoscopes and Macintosh Laryngoscope for Tracheal Intubation in Simulated Out-of-Hospital Conditions: A Randomized Crossover Manikin Study"

_healthcare, 2024, doi:10.3390/healthcare12040452_

Round 1

Reviewer 1 Report

Comments and Suggestions for Authors

Author Response

Reviewer #1

Improvement Sugestion

  • Conducting similar studies with experienced practitioners could provide more insights.
  • Testing in real-life emergency scenarios to validate findings.
  • Expanding the study to include different patient profiles and conditions.
  • The rows 235 and 272 contain some red text which raises the question if this is the final version of the article. - corrected
  • The limitation that the authors mention is that the number of participants is too low, but they also mention that the calculated number of participants is 30, which makes the number of participants not to be a limitation of this study - we estimated basing od published papers, that 50 intubation is required to achieve skills in intubation (Anaesth Intensive Care 2008; 36: 487-488)

We agree with all the reviewer's comments. All comments were included in the article.

Reviewer 2 Report

Comments and Suggestions for Authors

Congratulations on a well-designed study in a unique setting of patient population. The methods are well designed and explained in detail. This will provide some guidance for the use of video laryngoscopes in emergency out-of-hospital setting. Please see specific comments below.

Abstract: Abstract is too long and too many numbers and results included. Please restructure the abstract to provide a very brief introduction and the most important and relevant results.

Methods: head was placed in neutral position. were the participants allowed to extend the neck or was it kept neutral throughout the procedure to mimic real world scenario trauma patients?

p1 L27: and (22.75 sec. (IQR: 16.45-26.13) vs. 34.52 sec. (IQR: 27.62-39.15), p<0.001)- for which laryngoscope?

L44-45: n our study, the I-View and UESCOPE VL-400 video laryngoscopes provided better intubation results than the Macintosh laryngoscope. - please explain this conclusion is based on what aspect of the results? is this clinically significant and worth the extra cost of these laryngoscopes?

L 100-107: does the instructor demonstration followed each manufacture recommended techniques? for example, how did you use airtraq device for intubation- manufacture recommended technique does not include using a style for intubation with airtraq and if you place the ETT in the airtraq tube channel, it will be very difficult to use a stylet- does this affect your study results (airtraq has the lowest success rate- was it used as per recommended technique?)

166-167: 30 seconds was considered to be a significant difference for power calculation purposes. none of the devices had a 30 sec mean difference- based on this power calculation, is it possible for this study to conclude that one device was better than the others when the difference in TI is only less than 10 seconds?

175- 6 had performed more than 20 intubations in manikin- does this still be considered little to no experience? anyone had experience with actual patients prior to study?

185- similar to abstract- Is this for airtraq device?

186-187: without stylet is 16.66 sec and with stylet is 20.3 sec- change the phrase or flip the numbers.

Discussion:

223: For secondary end points, it also proved to be user friendly- based on what results? both I-view and UESCOPE have similar results and p values are > 0.05

235: please double check this sentence

265: POGO? expand

272-273: please double check the sentences.

Author Response

Comments:

Abstrect: Abstract: Abstract is too long and too many numbers and results included. Please restructure the abstract to provide a very brief introduction and the most important and relevant results.

Answer: We agree with all the reviewer's comments.

Line 27: deleted values (IQR: 17.28-26.77),

Line 28: deleted values (IQR: 19.4-36.06) and (IQR: 14.31-25.13)

Line 29: deleted values (IQR: 16.37-37.65), (IQR: 16.45-26.13) and (IQR: 27,62-

Line 29: term Airtraq was added

Line 30 delated value – 39.15)

Line 31: delated values (16.66 sec(IQR: 13.91-21.25) vs. 20.3 sec (IQR: 14.9-24.6), p=0,213)

Line 34: delated value (p=1.000)

Line 35: delated value (p=0.056)

Line 37: delated values (p=0.241 and p=0,165; and word ,,respectively” was removed

Line 40: delated values (p<0,001) twice

Line 41: delated value (p=0.240)

Line 42: term removed – and p=0.132, and word ,,respectively” was removed

Line 43: delated values (IQR:2-3) and (IQR:2-3) and (IQR:2-3)

Line 44: deleted value (IQR:2-3)

Line 45 and 46: added sentence - in terms of time needed to intubate, glottis visibility and reduction in dental damage.

Comments:

Methods: head was placed in neutral position. were the participants allowed to extend the neck or was it kept neutral throughout the procedure to mimic real world scenario trauma patients?

Answer:

 It kept neutral throughout the procedure to mimic real world scenario trauma patients.

Line 110 added sentence – with the manikin's head in neutral position.

Comments:

p1 L27: and (22.75 sec. (IQR: 16.45-26.13) vs. 34.52 sec. (IQR: 27.62-39.15), p<0.001)- for which laryngoscope?

Answer:

It's about the Airtraq laryngoscope

Line 29 added term – Airtraq

Comments:

L44-45: n our study, the I-View and UESCOPE VL-400 video laryngoscopes provided better intubation results than the Macintosh laryngoscope. - please explain this conclusion is based on what aspect of the results? is this clinically significant and worth the extra cost of these laryngoscopes?

Answer:

Line 45 and 46: added sentence - in terms of time needed to intubate, glottis visibility and reduction in dental damage.

Comments:

L 100-107: does the instructor demonstration followed each manufacture recommended techniques? for example, how did you use airtraq device for intubation- manufacture recommended technique does not include using a style for intubation with airtraq and if you place the ETT in the airtraq tube channel, it will be very difficult to use a stylet- does this affect your study results (airtraq has the lowest success rate- was it used as per recommended technique?)

Answer:

In our study, Non-Chaneled Airtraq videalaryngoscope was used. In this case, the use of stylet to facilitate the insertion of the ETT into the patient's airway is not contraindicated by the manufacturer.

The term Non-Channeled Airtraq has been added to the title of the article and in the methodology section in description of studied devices.

Line 105,106 added sentence – based on manufacturer's recommendations

Line 117: term changed from – unchanneled to – non-channeled

Comments:

166-167: 30 seconds was considered to be a significant difference for power calculation purposes. none of the devices had a 30 sec mean difference- based on this power calculation, is it possible for this study to conclude that one device was better than the others when the difference in TI is only less than 10 seconds?

Answer:

In our study, we used a 30-second difference as the basis for our sample size calculation. This value was chosen to minimize the risk of identifying falsely significant differences, thereby reducing the potential for a lower p-value. Opting for a smaller difference would necessitate a larger sample size, which in turn, could artificially lower the p-values. Despite this strategy, and our use of non-parametric tests complemented by appropriate post-hoc analyses, we observed statistically significant differences between approaches that demonstrated smaller time differences. The implications of these differences are best assessed by experienced anesthesiologists rather than statisticians. Below is an anesthesiological commentary on this matter: "Based on our extensive experience, reducing intubation time by every 10 seconds is crucial. There are several reasons behind this: it enhances the comfort of the anesthesiologist and facilitates the administration of further anesthesia. Moreover, it reduces the risk of hypoxia in the brain, particularly in emergency patients.".

Comments:

175- 6 had performed more than 20 intubations in manikin- does this still be considered little to no experience? anyone had experience with actual patients prior to study?

Answer:

Based on the literature, we believe that people who have performed 20 manikin intubations still have little experience in manikin intubation. Only after performing more than 50 manikin intubations can they be said to have experience in intubation (Weller J, Segal R. The acquisition of airway skills by new trainee anaesthetists. Anaesth Intensive Care 2008; 36: 487-488.)

Line 182, 183: added sentence – No participant had an experience of more than 50 intubations on manikin model nor intubated a real patient.

Comments:

185- similar to abstract- Is this for airtraq device?

Answer:

It's about the Airtraq laryngoscope

Line 193 added term – Airtraq

Comments:

186-187: without stylet is 16.66 sec and with stylet is 20.3 sec- change the phrase or flip the numbers.

Answer:

Line 195: sentence changed from – with or without stylets to – without or with stylets

Comments:

Discussion:

223: For secondary end points, it also proved to be user friendly- based on what results? both I-view and UESCOPE have similar results and p values are > 0.05

Answer:

Line 232, 233, 234: sentence changed from – For secondary endpoints, it also proved to be highly user friendly with high comfort and low intubation effort to – For secondary endpoints, it also proved to be similar to UESCOPE in terms of comfort and intubation effort.

Comments:

235: please double check this sentence

Answer:

sentence changed from – at floor level, the absence of the need to maintain this line is important ???, to – at floor level, the absence of the need to maintain eye-glottis line is important.

Comments:

265: POGO? Expand

Answer:

Line 274,275: added sentence – (scale of assessment of visibility of glottis based on percentage of glottis opening)

Comments:

272-273: please double check the sentences.

Answer:

Line 282, 283: sentence changed from – when a stylet was used and worse than the best device??? when a stylet was not used, to – when a stylet was used.